# *Peel-1* negative selection promotes screening-free CRISPR-Cas9 genome editing in *Caenorhabditis elegans*

**Troy A. McDiarmid**[1], **Vinci Au**[2], **Donald G. Moerman**[2], **Catharine H. Rankin**[1,3]*

**1** Djavad Mowafaghian Centre for Brain Health, University of British Columbia, Vancouver, British Columbia, Canada, **2** Department of Zoology, Life Sciences Centre, University of British Columbia, Vancouver, British Columbia, Canada, **3** Department of Psychology, University of British Columbia, Vancouver, British Columbia, Canada

* crankin@psych.ubc.ca

## Abstract

Improved genome engineering methods that enable automation of large and precise edits are essential for systematic investigations of genome function. We adapted *peel-1* negative selection to an optimized Dual-Marker Selection (DMS) cassette protocol for CRISPR-Cas9 genome engineering in *Caenorhabditis elegans* and observed robust increases in multiple measures of efficiency that were consistent across injectors and four genomic loci. The use of Peel-1-DMS selection killed animals harboring transgenes as extrachromosomal arrays and spared genome-edited integrants, often circumventing the need for visual screening to identify genome-edited animals. To demonstrate the applicability of the approach, we created deletion alleles in the putative proteasomal subunit *pbs-1* and the uncharacterized gene *K04F10.3* and used machine vision to automatically characterize their phenotypic profiles, revealing homozygous essential and heterozygous behavioral phenotypes. These results provide a robust and scalable approach to rapidly generate and phenotype genome-edited animals without the need for screening or scoring by eye.

## Introduction

Genome engineering–the ability to directly manipulate the genome, is a powerful approach to investigate its encoded functions. The nematode *Caenorhabditis elegans* has a rich history as a pioneering model system for the development of increasingly sophisticated methods to engineer the genome [1, 2]. For decades, genome engineering in *C. elegans* relied on random mutagenesis to induce mutations or integrate transgenes, which often resulted in unwanted background mutations, transgene silencing, or overexpression [1, 2]. The development of *Mos1* transposon-mediated Single Copy Insertion (MosSCI) and Deletion (mosDEL) finally allowed for deletion or insertion of designer sequences at a single copy into defined locations in the genome [1, 3–6]. While immensely impactful, this method was limited in that it required the availability of a transposon at the edit site, preventing many edits from being made at the desired locus. Zinc finger and transcription activator-like nucleases offered more specificity

**Data Availability Statement:** Strains and reagents are available from the CGC or upon request. All raw and processed data underlying the results presented in the study are available at: (https://doi.org/10.5683/SP2/FVEEWE), All analysis code and

the results of all statistical tests and are available at: (https://github.com/troymcdiarmid/peel-1).

**Funding:** This work was supported by a Canadian Institutes of Health Research (CIHR) (https://cihr-irsc.gc.ca/e/193.html) Doctoral Award to TAM, a CIHR grant to DGM (PJT-148549), a NIH grant 5R24OD023041 to DGM, and a CIHR project grant (PJT-165947) to CHR. The funders had no role in study design, data collection and analysis, decision to publish, or preparation of the manuscript.

**Competing interests:** The authors have declared that no competing interests exist.

but required considerable design effort for each new target, hindering their widespread adoption [7–9]. The discovery of CRISPR bacterial immune systems, followed promptly by their repurposing as a relatively easy to program RNA-guided system to target various effector domains (most notably the Cas9 nuclease) to precise locations in the genome revolutionized genome engineering across model systems [10–14]. In the short time since it's development, the versatility of CRISPR-based systems have allowed a remarkably diverse array of edits and modifications to be made in *C. elegans*, from single nucleotide variants and indels to larger deletions, insertions, direct replacements of entire genes and even programmed chromosomal rearrangements [2, 9, 15–27].

Despite these remarkable advances, CRISPR-based approaches for genome engineering in *C. elegans* still face major challenges that limit their efficiency and thus the scale and complexity of projects that can be achieved in practice. These can be broken down into issues that directly limit genome editing efficiency (e.g. the efficiency of Cas9 inducing a DNA double strand break or homology directed repair) or obstacles in screening that impede the identification and recovery of genome-edited animals. Two major challenges in screening are that: 1) following microinjection of transgene DNA many of the $F_1$ progeny of injected $P_0$ adults will not be transgenic, and 2) even among transgenic animals, genome-edited animals (or integrants) are rare relative to the number of animals harboring transgenes as extrachromosomal arrays (referred to hereafter as 'arrays'). Both of these factors severely complicate the recovery of *bona fide* genome-edited animals.

To begin to address these challenges, multiple selection schemes for CRISPR genome editing have been developed [1, 2]. These include approaches based on editing of a secondary locus with a visible phenotype (e.g. co-CRISPR), or progressively more elaborate positive and negative selection schemes to enrich for animals where genome editing has occurred (reviewed in [2]). Two recent approaches coupled visual markers with drug resistance genes housed in Cre recombinase-excisable cassettes, allowing for drug selection against non-transgenic progeny [28, 29]. These selection cassette methods effectively solved one of the major limitations of CRISPR genome editing in *C. elegans* by killing virtually all non-transgenic animals and provided a means to visually differentiate integrants from arrays, and thus represent the dominant methods for complex edits today. However, these approaches still face the limitation that animals harboring extrachromosomal arrays will also be resistant to drug selection and outnumber the desired integrants, necessitating cumbersome manual screening and isolation of putative genome-edited animals. Thus, there is great need for an approach that simultaneously selects against both non-transgenics and arrays, leaving only genome-edited animals.

PEEL-1 is a naturally occurring *C. elegans* sperm-derived toxin that is normally counteracted in the embryo by its antidote, ZEEL-1. Importantly, ectopic expression of *peel-1* at later life stages causes cell death and lethality [30]. This discovery motivated repurposing *peel-1* for array negative selection, in which a plasmid encoding heat shock driven *peel-1* is used to kill animals harboring arrays, thereby enriching for genome-edited integrants who have since lost the toxic array. Interestingly, while *peel-1* negative selection was developed as a component of MosSCI [4], and has been used in early CRISPR methods prior to the advent of excisable selection cassettes [20, 26], it's use has faded in recent years.

Here, we integrate *peel-1* negative selection with an optimized CRISPR-Cas9 genome editing protocol for combined negative selection against both non-transgenics and arrays [25]. This scheme is built on the Dual-Marker Selection (DMS) cassette method [28], which does not use heat shock driven cre-recombinase to excise it's selection cassette, making this approach compatible with heat shock driven *peel-1*. The combination of optimized guide selection, Cas9 RiboNucleoProtein (RNP) complexes, antibiotic selection for transgenics and *peel-1* negative selection against arrays effectively enriched for integrant animals to the point that

they would often take over culture plates, allowing for screening-free genome editing. We then applied our approach to generate deletion alleles in the putative proteasome subunit *pbs-1* and the uncharacterized gene *K04F10.3* to investigate the functional roles of these genes using machine vision. By combining our genome editing approach with automated machine vision phenotyping, we demonstrate the feasibility of generating and phenotyping genome-edited animals without the need for manual screening or scoring, opening the door to systematic investigations of genome function.

## Results

### An optimized peel-1-DMS pipeline effectively kills arrays and spares genome-edited integrants

We used an optimized DMS genome editing strategy and guide selection tool (http://genome.sfu.ca/crispr/) to design programmed deletions at two separate loci, the uncharacterized genes *F53B6.7* and *F10E9.2* (**Fig 1A**) [25, 28]. These deletions are predicted to result in null alleles. In this DMS strategy, guide RNAs are designed to induce a DNA double-strand break at the target locus (**Fig 1A**). These guides are injected along with Cas9 and a cocktail of plasmids including a homology directed repair template as well as pharyngeal (*Pmyo-2::mCherry*) and body wall muscle (*Pmyo-3::mCherry*) fluorescent co-injection markers (**Fig 1A**). The repair template consists of homology arms corresponding to the regions upstream and downstream of the cut site/intended edit site (e.g. in the case of a deletion the homology arm boundaries border the region to be deleted) as well as a dual-marker selection cassette housed between loxP sites (**Fig 1A**). The dual-marker selection cassette consists of a *Pmyo-2::GFP* pharyngeal fluorescent GFP

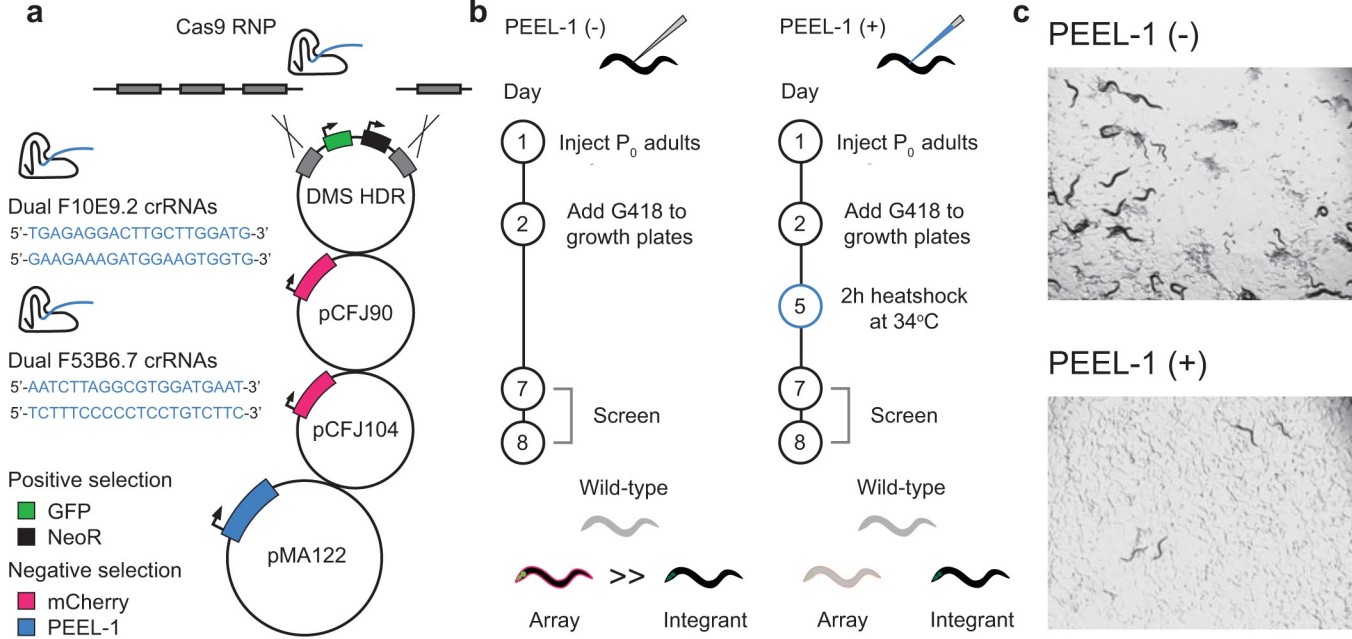

**Fig 1. An optimized peel-1-DMS CRISPR-Cas9 genome editing pipeline kills arrays and spares genome-edited integrants.** (A) Schematic of the peel-1-DMS CRISPR-Cas9 genome editing method. Dual crRNAs targeting genes of interest are injected as RNPs in complex with Cas9 to induce double strand breaks. A homology-directed repair template is used to integrate a *myo-2::GFP* pharyngeal visual marker and a Neomycin resistance gene at the cut site for integrant positive selection. Co-injected extrachromosomal mCherry markers provide visual selection against arrays while *peel-1* negative selection kills animals harboring arrays. In the standard DMS method arrays are manually distinguished from arrays based on dimness/consistency of GFP expression in the pharynx. (B) Injection and selection protocols/experimental design to test the efficacy of peel-1-DMS selection compared to our previously reported DMS method. (C) Peel-1-DMS selection kills arrays while sparing genome-edited integrants. Images from the F53B6.7 experiment 7 days post-injection.

marker and a *Prps-27*::*neoR* antibiotic (Neomycin/G418) resistance gene. Upon successful induction of a DNA double strand break by Cas9 the repair template is integrated at the target locus in a subset of animals (**Fig 1A**). If desired, the loxP sites also allow for optional excision of the selection cassette via a second round of injection of a plasmid encoding Cre recombinase for near scarless genome editing.

To improve screening efficiency, the antibiotic Neomycin (G418) is applied for drug selection 24h after injection to kill virtually all non-transgenic $F_1$ progeny (**Fig 1B**). However, this strategy produces both *bona fide* genome-edited integrants and unwanted animals harboring the injected plasmids as extrachromosomal arrays (**Fig 1B, left**). Both of these populations will be resistant to Neomycin, necessitating further screening. Because extrachromosomal arrays are often variably overexpressed different pharyngeal cells of array-carrying animals will express variable amounts of the pharyngeal GFP marker. Integrants, in contrast, will express only a single copy of GFP from each genome-edited locus in every pharyngeal cell. This allows for visual differentiation of genome-edited integrants from animals harboring arrays–array-carrying animals display bright and uneven GFP that is absent in some cells while integrants display even and dim GFP signal in every pharyngeal cell. A proportion of genome-edited integrants will also lose the body wall and pharyngeal muscle mCherry markers present in the extrachromosomal arrays, allowing for another level of visual selection against arrays (for an in depth description of the DMS approach see [28] and [25]).

While conceptually appealing, this scheme normally results in far more arrays than integrants, which can only be differentiated manually by screening for the presence of body wall and pharyngeal muscle mCherry markers and/or the brightness/consistency of GFP fluorophore expression in the pharynx. In practice the mCherry transgenes are often too dim to confidently visualize, leaving GFP as the only way to identify integrants, effectively reducing the process to searching for a dim needle in a variably bright haystack (**Fig 1B**). We hypothesized that the addition of *peel-1* negative selection to the optimized DMS pipeline (referred to hereafter as peel-1-DMS) delivered on the 5$^{th}$ day following injection would kill arrays without killing genome-edited integrants, which by that point would have lost the toxic extrachromosomal array. Indeed, we observed that while there were several integrants that had survived selection on days 7 and 8 post-injection, heat shock induction of *peel-1* killed arrays (**Fig 1C**).

## Peel-1-DMS attenuates array-driven overpopulation/starvation and promotes screening-free genome editing at diverse loci

An important limitation of the standard DMS protocol is that worms carrying arrays crowd the culture plates, rapidly exhausting the food source and starving the population (**Fig 1C, top panel**). This prevents the rare integrants from surviving and reproducing, making screening more difficult. A single heat shock to induce *peel-1* negative selection decreased overpopulation-induced plate starvation approximately two-fold across two injectors each targeting the two *F53B6.7* and *F10E9.2* loci (**Fig 2A**). Although a potential concern might be that *peel-1* would also kill array-carrying integrants, effectively decreasing the editing efficiency/recovery of genome-edited animals, we observed no differences in the efficiency (or total number of integrant animals retrieved) of CRISPR-Cas9 genome editing following *peel-1* treatment (**Fig 2B**). We also did not observe an increase in male progeny following *peel-1* induction (potentially because most animals that could be males would be arrays and have been killed off and/ or a single 2h 34°C heat shock is insufficient to cause nondisjunction of the X chromosome and increase male proportions to a noticeable degree at a population level). Most importantly, the combined selection against arrays and reduced plate starvation allowed the integrants to win out, resulting in "pure" integrant plates and removing the need for any screening. We

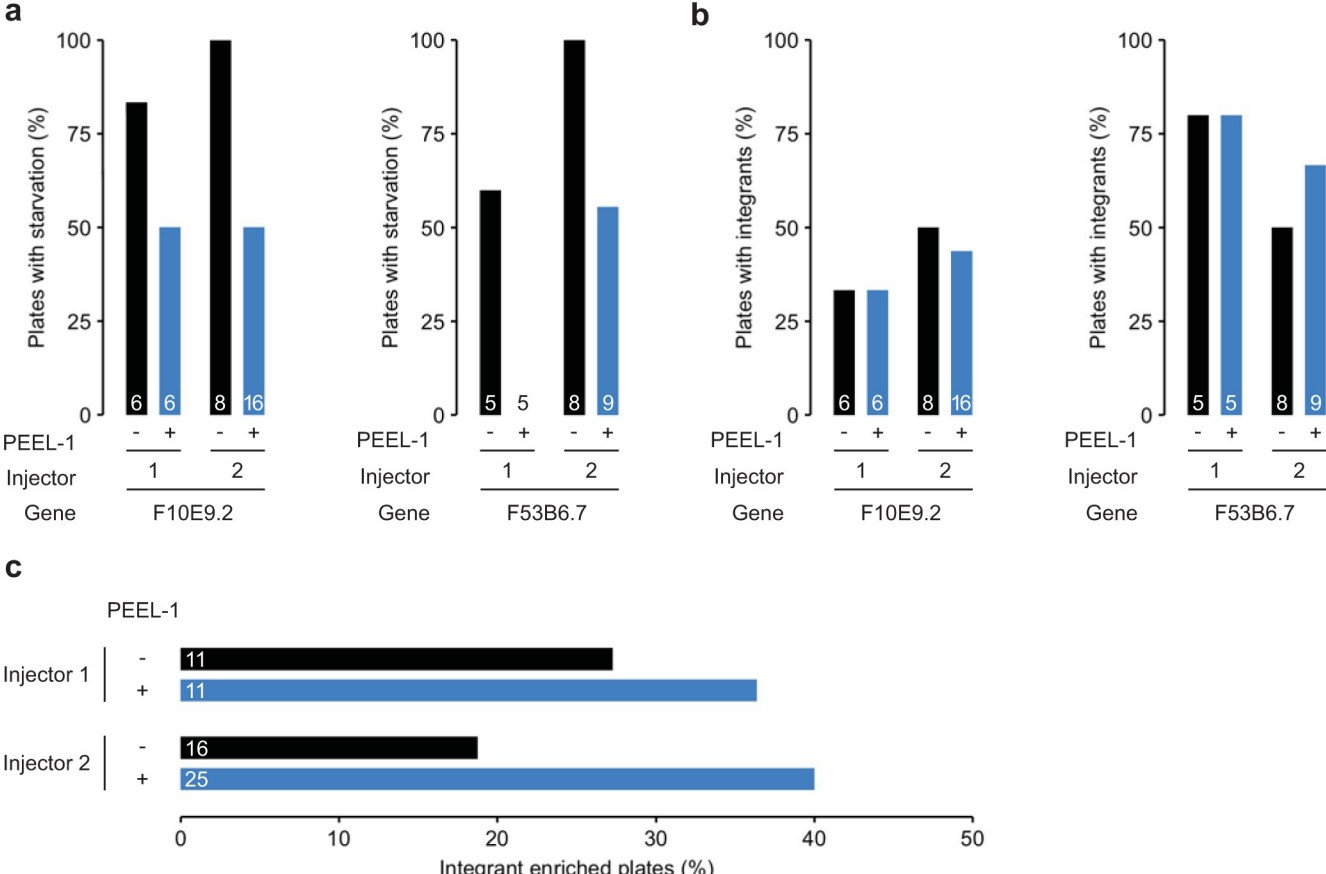

**Fig 2. Peel-1-DMS attenuates array-driven overpopulation/starvation and promotes screening-free genome editing at diverse loci.** (A) Proportion of injected plates showing signs of starvation with or without *peel-1* negative selection on the 7th day post-injection. Peel-1-DMS selection resulted in robust reductions in starvation across two injectors each targeting two distinct genomic loci. Note that the absence of a bar indicates that no plates in that group showed signs of starvation. For the *F10E9.2* target, n = 6 plates for *peel-1* (-) and 6 plates for *peel-1* (+) for injector 1, and n = 8 plates for *peel-1* (-) and 16 plates for *peel-1* (+) for injector 2. For the *F53B6.7* target, n = 5 plates for *peel-1* (-) and 5 plates for *peel-1* (+) for injector 1, and n = 8 plates for *peel-1* (-) and 9 plates for *peel-1* (+) for injector 2. Note that each independent plate consists of 4 injected $P_0$ worms. (B) Peel-1-DMS selection did not alter the proportion of plates from which integrants were recovered (the number of plates per condition is the same as in panel A). (C) Proportion of plates enriched for integrant animals 11 (F10E9.2) or 12 (F53B6.7) days post-injection. For injector 1, n = 11 plates for *peel-1* (-) and 11 plates for *peel-1* (+). For injector 2, n = 16 plates for *peel-1* (-) and 25 plates for *peel-1* (+). Note that each independent plate consists of 4 injected $P_0$ worms. Peel-1-DMS selection robustly increased the proportion of integrant enriched plates.

observed robust enrichment for pure integrant plates that was consistent across genomic loci (**Fig 2C**). Thus, through the addition of *peel-1* negative selection to an optimized DMS cassette method, *C. elegans* genome engineers can recover integrants from diverse loci without visual screening.

## Peel-1-DMS is effective at two additional loci and multiple heat shock rounds further increases killing

While peel-1-DMS is highly effective at killing arrays, some animals do escape negative selection. However, animals that are not genome edited and managed to escape both *peel-1* and drug selection should still harbor the toxic *peel-1*-containing array, suggesting a testable hypothesis that additional heat shock rounds to induce *peel-1* at later time points would further increase killing. To simultaneously test this hypothesis and validate our approach at additional target loci, we used peel-1-DMS to generate deletion alleles in two additional genes, the

putative proteasome subunit *pbs-1* and the uncharacterized gene *K04F10.3*, and subjected animals to multiple heat shock rounds on either days 5 (standard), 5 & 7, or 5, 7, & 9. We then measured the strength of negative selection by scoring for signs of array-driven starvation 12 days post-injection. We did not observe increased selection with multiple heat shock rounds for *pbs-1*, as a single heat shock induction of *peel-1* on day 5 killed enough array carrying animals to prevent any signs of starvation for the entire 12-day testing period (**Fig 3A**). However, we did observe increased selection following multiple heat shock rounds for the *K04F10.3* target, indicating that while a single 2h heat shock at 5d is sufficient for effective selection, multiple heat shock rounds further increased killing using peel-1-DMS (**Fig 3B**). Thus, for both targets peel-1-DMS attenuated array overpopulation/plate starvation and simplified recovery of genome-edited animals from integrant enriched plates. Taken together, these results demonstrate that peel-1-DMS is effective at multiple additional loci and that additional heat shock rounds can further strengthen *peel-1* negative selection.

## Machine vision phenotypic profiles for novel deletion alleles of *pbs-1* and *K04F10.3* generated via peel-1-DMS

Peel-1-DMS allows for isolation of genome-edited animals without the need for manual screening. If paired with automated phenotypic characterization, this would open to the door to full automation of diverse investigations of genome function in *C. elegans*. Toward this goal, we used our automated machine vision phenotyping system, the Multi-Worm Tracker (**Fig 4A**) [31] to generate phenotypic profiles for the *pbs-1* and *K04F10.3* deletion mutants we generated with peel-1-DMS. *pbs-1* was previously associated with a high-penetrance embryonic lethal phenotype in genome-wide RNAi screens [32–34]. We confirm these knockdown results with precise CRISPR-Cas9 deletion alleles, ruling out compensation and definitively designating this gene as essential under standard laboratory growth conditions. Importantly, the DMS cassette does not use morphology or behavior altering selection markers and includes an easily visualized dominant pharyngeal GFP marker–allowing for simplified maintenance and behavioral analysis of strains harboring heterozygous knockouts in essential genes, such as *pbs-1*.

High-throughput phenotypic profiling using the Multi-Worm Tracker revealed that *K04F10.3* mutant worms were shorter in length but not width, and displayed a more kinked

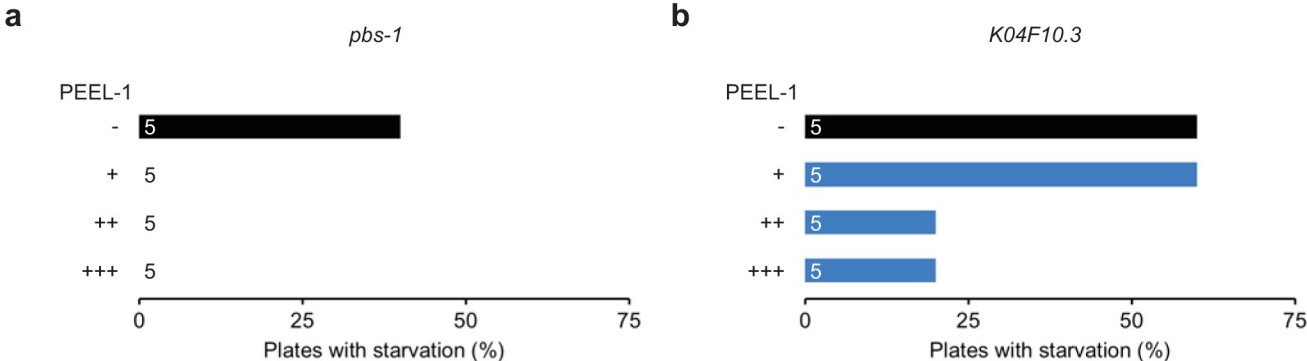

**Fig 3. Peel-1-DMS is effective at two additional loci and multiple heat shock rounds further increases killing.** (A) Proportion of injected plates targeting *pbs-1* showing signs of starvation on the 12th day post-injection following one or multiple heat shock rounds to induce *peel-1*. (-) = no *peel-1* treatment, (+) = a single heat shock round on day 5, (++) = Two heat shock rounds on days 5 and 7, (+++) = three heat shock rounds on days 5, 7, and 9. Note that the absence of a bar indicates that no plates in that group showed signs of starvation. (B) Proportion of injected plates targeting *K04F10.3* showing signs of starvation on the 12th day post-injection following multiple heat shock rounds to induce *peel-1*. (-) = no *peel-1* treatment, (+) = a single heat shock round on day 5, (++) = Two heat shock rounds on days 5 and 7, (+++) = three heat shock rounds on days 5, 7, and 9. N = 5 independent plates for all conditions. Note that each independent plate consists of 4 injected $P_0$ worms.

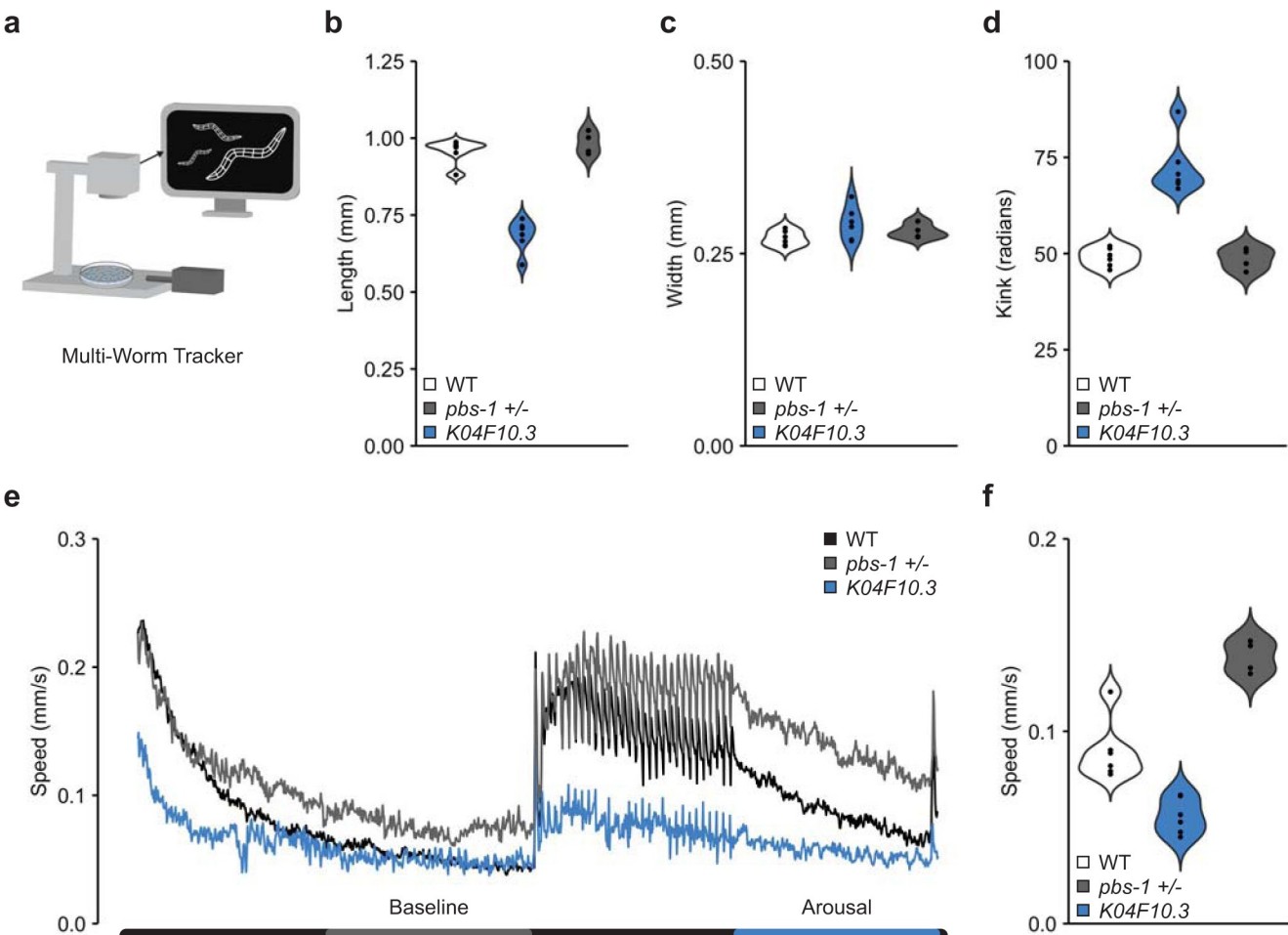

**Fig 4. Deletion alleles of *pbs-1* and *K04F10.3* generated via peel-1-DMS selection reveal homozygous lethal and heterozygous behavioral phenotypes.**
(A) Schematic of the Multi-Worm Tracker machine vision phenotyping system. A high-resolution camera records a plate of worms while the Multi-Worm Tracker software creates comprehensive digital representations of the worms in real time from which multiple phenotypes are later computationally extracted offline. The Multi-Worm Tracker also coordinates delivery of stimuli, e.g. the mechanosensory stimuli delivered to the plates via a push solenoid used here. (B) Worm length across genotypes. Each dot represents the mean of an independent plate replicate. Each plate consists of 20–100 worms. (C) Worm width across genotypes. Each dot represents the mean of an independent plate replicate. Each plate consists of 20–100 worms. (D) The degree of body posture kink (in radians) across genotypes. Each dot represents the mean of an independent plate replicate. Each plate consists of 20–100 worms. (E) Mean absolute movement throughout the tracking session. *pbs-1* heterozygotes display initially normal locomotion speed and prolonged arousal following mechanosensory stimuli. (F) Quantification of aroused movement speed in the period following mechanosensory stimulation across genotypes. Each dot represents the mean of an independent plate replicate. Each plate consists of 20–100 worms. WT = PD1074 wild-type control, *K04F10.3* = *K04F10.3(gk5669)*, *pbs-1* +/- = *pbs-1(gk5673)/+*.

body posture than wild-type worms (**Fig 4B–4D**). (*K04F10.3* homozygous mutants displayed reduced fecundity but were still amenable to tracking). *pbs-1* heterozygotes, in contrast, displayed apparently normal morphology (**Fig 4B–4D**) and initial locomotion speed (**Fig 4E**), and like *K04F10.3* mutants exhibited generally normal habituation of reversal responses to repeated mechanosensory stimulation (**S1 Fig**). Interestingly however, detailed behavioral analyses revealed that *pbs-1* heterozygotes were more easily aroused by touch than wild-type worms, observed as a prolonged increase in movement speed following repeated mechanosensory stimulation (**Fig 4E and 4F**). Taken together, these results demonstrate that combining peel-1-DMS with machine vision phenotyping allows for the generation and characterization of genome-edited animals without the need for manual screening or scoring.

## Discussion

We developed an integrated peel-1-DMS CRISPR-Cas9 genome editing strategy and observed robust increases in multiple measures of efficiency. Peel-1-DMS selection dramatically reduced the number of arrays without altering the number integrants. We demonstrated the broad applicability of the approach by generating four deletion alleles in different genes and phenotyping two of the deletion alleles in *pbs-1* and *K04F10.3*, revealing homozygous lethal and heterozygous behavioral phenotypes. Combining peel-1-DMS genome engineering with automated phenotypic characterization represents a streamlined strategy to precisely edit and functionally annotate the genome without any need for visual screening or scoring, opening the door for large-scale knock-out and phenotyping efforts.

*Peel-1* negative selection was developed as a key component MosSCI and improved the efficiency of the technique [1, 4]. *Peel-1* negative selection was subsequently used in early laborious PCR screening-based CRISPR methods before the advent of excisable drug selection cassettes, but it's use faded shortly after. There are several potential reasons for this, including incompatibility with the Self-Excising Cassette (SEC) CRISPR method that uses a heat shock inducible Cre recombinase to excise the selection cassette from the genome–meaning the heat shock used to induce *peel-1* would also destroy the repair template by excising the selectable markers prior to successful recovery of genome-edited animals. Of note, *peel-1* selection has previously been suggested as a possible inclusion to the original DMS method by it's developers [28, 35, 36]. We demonstrate here that combining *peel-1* with an optimized sgRNA selection, homology-directed repair, and Cas9 RNP DMS approach yields increased efficiency. Indeed, the largest screening efficiency increases were observed with the most efficient guides, and are likely due to the combined use of *peel-1* with highly-effective Cas9 RNP complexes and optimized guide selection [19, 25]. Importantly, our results suggest that the more efficient the editing of a particular locus is, the more likely it is that peel-1-DMS can drive the population to pure integrants.

Placing *peel-1* under an inducible promoter that does not require heat shock (e.g. a drug inducible promoter via further refinement of the Q system) [37, 38] would make it compatible with SEC-based CRISPR engineering. This would also allow peel-1-DMS selection to be used when creating edits that result in heat sensitivity or other phenotypes incompatible with heat shock. Alternatively, given the speed of the current approach, the SEC constructs could simply be redesigned using the peel-1-DMS framework or used without heat shock (standard DMS screening) in the rare cases where target perturbation would cause severe heat sensitivity to the 2h 34°C exposure period.

Our plasmid-based peel-1-DMS selection approach is particularly useful for large-scale genome edits due to the increased efficiency conferred by the integrated DMS cassette (e.g. replacing *C. elegans* genes with genes from other species, deleting entire open reading frames, or deleting several adjacent genes simultaneously) [24, 25]. However, several methods based on linear DNA repair templates have recently been developed that allow for efficient generation of small edits (typically up to fluorophore-sized insertions) in *C. elegans* without the need for cloning [2, 22, 39–44]. While these approaches currently require manual screening/isolation and/or PCR validation to identify genome-edited animals and have lower efficiency for large edits, with further optimization, peel-1-DMS could be adapted to these strategies to allow cloning- and screening-free genome editing. In principle, this could be achieved through integration of the ~2 kilobase *Prps-27*::*neoR* Neomycin resistance sequence (e.g. via Nested CRISPR) [43] and simultaneous use of a *peel-1* negative selection plasmid to kill array-carrying animals.

Combined peel-1-DMS selection will also be particularly useful for large-scale projects designed to repeatedly edit the same locus (e.g. creating large allelic series), as researchers will

simply have to identify a guide efficient enough to ensure integrant enriched plates, bypassing the need for screening on each subsequent edit. Every guide RNA we designed with our selection tool resulted in >10% integrant enriched plates when tested with peel-1-DMS selection. Further, peel-1-DMS will reduce the need for new users to learn to distinguish arrays from integrants based on subtle fluorescence patterns. Even in the cases where screening is still required, researchers now only have to differentiate uneven array GFP from even integrant GFP signal in a much smaller pool of animals, removing the need for access to multiple fluorescence channels and making it physically easier to single integrants.

Peel-1-DMS selection provides the *C. elegans* community a robust, cheap, and easy to implement method to increase the efficiency of diverse CRISPR-Cas9 genome engineering projects. Our results and the rapid pace of CRISPR method development in *C. elegans* suggest that recovering and functionally characterizing genome-edited animals in a screening and scoring-free manner may soon be the norm.

# Materials and methods

## Strains and maintenance

Strains were maintained on NGM (nematode growth medium) plates seeded with the *Escherichia coli* strain OP50 according to standard experimental procedures [45]. Strains were maintained at 20˚C unless otherwise noted. PD1074, the Moerman lab derivative of N2 [46], was used for all CRISPR-Cas9 genome editing and behavioral experiments.

## CRISPR-Cas9 genome engineering

The *C. elegans specific* guide RNA selection tool (http://genome.sfu.ca/crispr/) was used to identify the F53B6.7, F10E9.2, *pbs-1*, and *K04F10.3* targeting crRNAs (dual guides for each target). The complete list of crRNAs can be found in **S1 Table**.

Gene-specific crRNAs and universal tracrRNAs, both ordered from Integrated DNA Technologies (IDT), were duplexed according to manufacturer's instructions then incubated with purified Cas9 protein (kindly provided by the lab of Dr. Geraldine Seydoux, Johns Hopkins University) to create RNPs for injection.

Homology directed repair constructs were designed and constructed according to the optimized DMS protocol as previously described [25, 28]. Briefly, homology arms flanking the region to be deleted (450 bp homology with 50 bp adapter sequences for Gibson assembly) were ordered as 500 bp gBlocks from Integrated DNA Technologies (IDT). Note that we have recently switched to using eBlocks from IDT for homology arms that are more cost-effective and ship faster. Repair template plasmids were assembled using the NEBuilder Hifi DNA Assembly Kit (New England BioLabs) to incorporate homology arms into the *loxP + Pmyo-2::GFP::unc-54 3'UTR + Prps-27::neoR::unc-54 3'UTR + loxP* dual-marker selection cassette vector (provided by Dr. John Calarco).

Standard DMS injection mixes consisted of 2.5 ng/μl pCFJ90 (*Pmyo-2::mCherry*), 5 ng/μl pCFJ104 (*Pmyo-3::mCherry*), 50 ng/μl gene-specific repair templates, and 0.5 μM gene-specific Cas9 RNPs. Peel-1-DMS injection mixes were prepared the same except that they included pMA122 (*Phsp16.41::peel-1*) at 10ng/μl for *peel-1* negative selection (**Fig 1A**). pCFJ90—*Pmyo-2::mCherry::unc-54utr* (Addgene plasmid # 19327; http://n2t.net/addgene:19327; RRID: Addgene_19327), pCFJ104—*Pmyo-3::mCherry::unc-54* (Addgene plasmid # 19328; http://n2t.net/addgene:19328; RRID:Addgene_19328), and pMA122—*peel-1* negative selection (Addgene plasmid # 34873; http://n2t.net/addgene:34873; RRID:Addgene_34873) were gifts from Dr. Erik Jorgensen.

Adult P$_0$ hermaphrodites were microinjected and then transferred in groups of 4 to standard culture plates to recover [25, 47]. 24h following microinjection 500μl of 25mg/ml G418 was added to the culture plates for antibiotic selection (**Fig 1**).

### *Peel-1* induction

Five days following microinjection, plates were transferred from 20˚C incubation to a 34˚C incubator for a 2h heat shock to induce *peel-1*. For experiments involving multiple heat shocks the same procedure was repeated on days 7 and 9.

### Screening and quantification

Genome-edited animals were identified by *peel-1* and/or G418 resistance, loss of extrachromosomal array markers, and uniform dim fluorescence of the inserted GFP.

Experimenters blinded to condition scored plates for signs of starvation (exhausted OP50 food source) at the indicated time points. Plates where virtually all animals (>95%) were putative integrants based on *peel-1* and/or antibiotic resistance and visual markers were counted as integrant enriched.

### Genotype confirmation

Correct insertion of the DMS cassette sequence was confirmed by amplifying the two regions spanning the upstream and downstream insertion borders using PCR. The genotyping strategy is essentially as described for deletion allele generation via DMS cassette insertion in [25] and [28].

Gene-specific forward primers were used with a universal reverse primer located within the GFP coding region of the DMS cassette: CGAGAAGCATTGAACACCATAAC to amplify the upstream insertion region for sequence confirmation.

Gene-specific reverse primers were used with a universal forward primer located within the Neomycin resistance gene of the DMS cassette: CGAGAAGCATTGAACACCATAAC to amplify the downstream insertion region for sequence confirmation.

Gene-specific wild-type primers were used in conjunction with either the forward or reverse gene-specific primer to detect partial/imperfect edits or gene duplications.

The complete list of all gene-specific forward and reverse sequence confirmation primers can be found in **S1 Table**.

### Strain list

The following strains were generated by CRISPR-Cas9 genome engineering via microinjection of plasmid DNA and Cas9 RNPs:

VC4544 F53B6.7(gk5615[+LoxP Pmyo-2::GFP::unc-54 UTR Prps-27::NeoR::unc-54 UTR LoxP+]) IV

VC4352 F10E9.2(gk5435[+LoxP Pmyo-2::GFP::unc-54 UTR prps-27::NeoR::unc-54 UTR LoxP+]) I

VC4353 F10E9.2(gk5436[+LoxP Pmyo-2::GFP::unc-54 UTR prps-27::NeoR::unc-54 UTR LoxP+]) I

VC4603 pbs-1(gk5673[+LoxP Pmyo-2::GFP::unc-54 UTR Prps-27::NeoR::unc-54 UTR LoxP+]) IV

VC4599 K04F10.3(gk5669[+LoxP pmyo-2::GFP::unc-54 UTR prps-27::NeoR::unc-54 UTR LoxP+]) I

Strains harboring programmed deletions in each of these four genes are available from the Caenorhabditis Genetics Center or upon request.

## Behavioral assays

For the Multi-Worm Tracker mechanosensory habituation paradigm animals were synchronized for behavioral testing on NGM plates seeded with 50 μl of OP50 liquid culture 12–24 hours before use. For PD1074 wild-type controls and *pbs-1* heterozygous deletion mutants five gravid adults were picked to plates and allowed to lay eggs for 3–4 hours before removal. Due to reduced fecundity, *K04F10.3* homozygous mutants were age synchronized either by allowing 25 gravid adults to lay eggs for 3–4 hours before removal or by dissolving 25 gravid adults on the tracking plates in bleach to liberate their eggs. Both bleaching- and egg laying-based synchronization produced consistent results for all analyses and so were pooled to a single genotype group representing *K04F10.3* mutants. 72h old *pbs-1* heterozygous deletion mutant adults were identified via pharyngeal GFP (homozygous mutants are lethal while wild-type homozygotes do not carry GFP) and transferred to fresh Multi-Worm Tracker plates and tracked 24h later. For all Multi-Worm Tracker experiments 4–6 plates (20–100 worms/plate) were run for each strain. The animals were maintained in a 20°C incubator for 96 hours prior to testing [48].

Our behavioral paradigm consisted of a 5-minute period to recover from being placed on the tracker followed by a 5 min baseline period from which we computed multiple measures of morphology and baseline locomotion (**Fig 4E**) [48]. Beginning at 10 minutes we administered 30 mechanosensory stimuli to the Petri plate holding the animals at a 10 second interstimulus interval (ISI) using an automated push solenoid (**Fig 4A**). *C. elegans* respond to a mechanosensory stimulus by emitting a reversal response (crawling backwards) allowing us to assess multiple measures of naïve sensitivity (e.g. reversal likelihood, duration, etc.; **S1 Fig**). With repeated stimulation there is a decrease in the likelihood of a reversal, as well as the duration, speed, and distance of reversals (habituation learning; **S1 Fig**). Following habituation training, we allowed a 5-minute recovery period after which we administered a 31$^{st}$ stimulus to gauge spontaneous recovery from short-term habituation—an assay of short-term memory retention [48].

## Multi-Worm Tracker behavioral analysis and statistics

Multi-Worm Tracker software (version 1.2.0.2) was used for stimulus delivery and image acquisition. Phenotypic quantification with Choreography software (version 1.3.0_r103552) used "—shadowless", "—minimum-move-body 2", and "—minimum-time 20" filters to restrict the analysis to animals that moved at least 2 body lengths and were tracked for at least 20 s. Standard choreography output commands were used to output morphology and baseline locomotion features [31]. A complete description of the morphology, baseline locomotion, sensory, and habituation learning features can be found in the Multi-Worm Tracker user guide (https://sourceforge.net/projects/mwt/) [31]. The MeasureReversal plugin was used to identify reversals occurring within 1 s (d$t$ = 1) of the mechanosensory stimulus onset. Comparisons of "final response" comprised the average of the final three stimuli. Arousal was defined as the increased mean absolute movement speed in the period following mechanosensory stimulation and prior to the delivery of the spontaneous recovery stimulus (**Fig 4E and 4F**; 600–1189 seconds). Custom R scripts organized and summarized Choreography output files [48]. No blinding was necessary because the Multi-Worm Tracker scores behavior objectively.

Phenotypic features were pooled across plate replicates for each mutant strain and means were compared to the mean of the wild-type distribution with an unpaired t-test implemented using a linear model in R with a Benjamini-Hochberg control of the false discovery rate at 0.001 [48]. Sample sizes for each behavioral assay were chosen to be either equal to or greater than sample sizes reported in the literature that were sufficient to detect biologically relevant differences. Final figures were generated using the ggplot2 package in R [49].

## Supporting information

**S1 Fig. Habituation of reversal responses to mechanosensory stimuli in *K04F10.3* homozygous and *pbs-1* heterozygous deletion mutants.** (A) Habituation of reversal probability across genotypes. (B) Habituation of reversal duration across genotypes. (C) Habituation of reversal speed across genotypes. Both *K04F10.3* homozygous and *pbs-1* heterozygous mutants displayed generally normal habituation (learned decrement) of reversal responses to repeated mechanosensory stimuli. Note that speed is consistently lower *K04F10.3* homozygous mutants, likely due to their reduced size (**Fig 4E**). Dots represent the mean of plate replicates (n = 4–6 plates per genotype). Each plate consists of 20–100 worms. Error bars represent standard error of the mean. WT = PD1074 wild-type control, *K04F10.3* = K04F10.3(gk5669), *pbs-1* +/- = *pbs-1(gk5673)/+*.
(EPS)

**S1 Table. crRNA and PCR primer sequences.**
(XLSX)

## Acknowledgments

We would like to thank Dr. John Calarco, Dr. Christian Frøkjær-Jensen, Dr. Geraldine Seydoux, Dr. Erik Jorgensen, and their labs for sharing constructs, reagents, and protocols or making them available. We would also like to thank members of the Rankin and Moerman labs for helpful discussions about the project.

## Author Contributions

**Conceptualization:** Troy A. McDiarmid, Vinci Au, Donald G. Moerman, Catharine H. Rankin.

**Data curation:** Troy A. McDiarmid, Vinci Au.

**Formal analysis:** Troy A. McDiarmid.

**Funding acquisition:** Donald G. Moerman, Catharine H. Rankin.

**Investigation:** Troy A. McDiarmid, Vinci Au, Donald G. Moerman, Catharine H. Rankin.

**Methodology:** Troy A. McDiarmid, Vinci Au, Donald G. Moerman, Catharine H. Rankin.

**Project administration:** Vinci Au, Donald G. Moerman, Catharine H. Rankin.

**Resources:** Donald G. Moerman, Catharine H. Rankin.

**Software:** Troy A. McDiarmid.

**Supervision:** Donald G. Moerman, Catharine H. Rankin.

**Validation:** Troy A. McDiarmid, Vinci Au.

**Visualization:** Troy A. McDiarmid.

**Writing – original draft:** Troy A. McDiarmid.

**Writing – review & editing:** Troy A. McDiarmid, Vinci Au, Donald G. Moerman, Catharine H. Rankin.

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
