## [Decision Letter · Decision Letter 0]

17 Jun 2020

PONE-D-20-16105

Peel-1 negative selection promotes screening-free CRISPR-Cas9 genome editing in Caenorhabditis elegans.

PLOS ONE

Dear Dr. Rankin,

Thank you for submitting your manuscript to PLOS ONE. After careful consideration, we feel that it has merit but does not fully meet PLOS ONE’s publication criteria as it currently stands. Therefore, we invite you to submit a revised version of the manuscript that addresses the points raised during the review process.

As this work present a a combination of technical strategies that were previously described elsewhere it is essential that all aspects of the method are thoroughly described, referenced and discussed within the manuscript. Please follow the indications from the reviewers to improve the manuscript with that objective in mind. 

We look forward to receiving your revised manuscript.

Kind regards,

Denis Dupuy, Ph.D.

Academic Editor

PLOS ONE

Reviewers' comments:

Reviewer's Responses to Questions

**Comments to the Author**

1. Is the manuscript technically sound, and do the data support the conclusions?

Reviewer #1: Yes

Reviewer #2: Yes

2. Has the statistical analysis been performed appropriately and rigorously? 

Reviewer #1: Yes

Reviewer #2: Yes

3. Have the authors made all data underlying the findings in their manuscript fully available?

Reviewer #1: Yes

Reviewer #2: Yes

4. Is the manuscript presented in an intelligible fashion and written in standard English?

Reviewer #1: Yes

Reviewer #2: Yes

5. Review Comments to the Author

Reviewer #1: In this manuscript the authors use negative selection based on expression of peel-1 to eliminate array carrying animals in the selection procedure for CRISPR/Cas9 integrants. Differentiating between array carrying animals and true integrants is a time-consuming step in genome editing procedures that use plasmid-based repair templates, and the use of peel-1 should indeed help speed up the procedure.

The manuscript data appears sound, but the manuscript would benefit from the inclusion of more details:

- The DMS editing strategy should also be summarized and/or visualized better. I had to go back to the original publication to look up the details. For example, it is true that the brightness and consistency of GFP fluorescence can be used to differentiate integrants from arrays, but it is not clear at all why that is. A slightly extended summary of the method would help the reader of this manuscript.

- The injected plasmids are also presented with very little detail. They are available online of course, but it would be convenient to have at hand.

- The paper could also benefit from a more extensive description of various numbers, like the number of animals injected, the number of plates, the number of injected animals per plate, and the number of line obtained. The numbers are there, but spread throughout the text, legends and methods.

Finally, the authors do not mention anywhere that peel-1 selection is not applicable for editing approaches that do not use plasmids as repair template.

Reviewer #2: Despite many germ nuclei can the targeted in a single injection with CRISPR reagents in C. elegans, the efficacy of CRISPR gene editing is still rather low, and therefore several methods have been developed to select animals carrying the edit of interest.

In this manuscript, the authors combine previous strategies to develop a new methodology.

In brief, to use resistance to antibiotics (in this case neomycin) as a marker of a CRISPR-Cas9 induced gene disruption, since the repair template is a plasmid, there is some residual activity from the no integrated plasmid that provides resistance to the antibiotic. To eliminate those worms carrying the extrachromosomal arrays made by plasmids, they add to the plasmid mix a plasmid expressing (upon heat shock) the toxin PEEL-1. In that way, they got plate enriched in worms with the right deletion (resistant to antibiotic because the insertion disrupting the gene, and not because resistance provided from the plasmid.

Strategies are not novel but it is a smart application of the existing tools. C. elegans is a pioneer in the applications of CRISPR technologies because of its genetic properties and short life cycle, and therefore any study on this matter would be of interest. The experiments are performed correctly and the manuscript is clear in its main messages. Thus, I think it would be suitable for publication in Plos One. Still, I have some comments and a few recommendations to improve the paper (mainly the discussion).

Producing mutations is very efficient in C. elegans with cloning-free CRISPR methods that would require not many PCRs to detect a mutant. In that sense, their approach is not of great interest to somebody with interest in producing a mutation by CRISPR. According to them, the interest of their methodology relies on “opening the door for genome-wide knock-out and phenotyping efforts” since the couple the CRISPR editing with an automated phenotyping device. Still, the main limitation for a large-scale experiment is the microinjection. They should mention this in the discussion and rather say that their approach would be valid to study gene families or a limited collection of genes, not for genome-wide approaches.

Order two 500 bp gBlocks, plus cloning, would cost money and time…particularly if they go for large-scale approaches. Such limitations would be bypassed with the insertion of a dsDNA with the rps-27 promoter plus the gene for neomycin resistance (I estimate 2Kb). This possibility should be discussed, and the articles from Dokshin et al 2018 (use Hybrid dsDNA donors) and Vicencio et al 2019 (Nested CRISPR) should be cited since they would provide a way to insert the 2Kb sequence that can provide resistance to the antibiotic and the time that a gene is mutated.

Also missing citation of Farboud et al, 2019, where authors comments on insertions of large fragments of DNA by CRISPR.

6. PLOS authors have the option to publish the peer review history of their article (what does this mean?). If published, this will include your full peer review and any attached files.

Reviewer #1: No

Reviewer #2: No

---

## [Author Response · Author response to Decision Letter 0]

28 Jul 2020

PONE-D-20-16105

Peel-1 negative selection promotes screening-free CRISPR-Cas9 genome editing in Caenorhabditis elegans.

PLOS ONE

Dear Dr. Rankin,

Thank you for submitting your manuscript to PLOS ONE. After careful consideration, we feel that it has merit but does not fully meet PLOS ONE’s publication criteria as it currently stands. Therefore, we invite you to submit a revised version of the manuscript that addresses the points raised during the review process.

As this work present a a combination of technical strategies that were previously described elsewhere it is essential that all aspects of the method are thoroughly described, referenced and discussed within the manuscript. Please follow the indications from the reviewers to improve the manuscript with that objective in mind. 

We look forward to receiving your revised manuscript.

Kind regards,

Denis Dupuy, Ph.D.

Academic Editor

PLOS ONE

Reviewers' comments:

Reviewer's Responses to Questions

Comments to the Author

1. Is the manuscript technically sound, and do the data support the conclusions?

Reviewer #1: Yes

Reviewer #2: Yes

2. Has the statistical analysis been performed appropriately and rigorously?

Reviewer #1: Yes

Reviewer #2: Yes

3. Have the authors made all data underlying the findings in their manuscript fully available?

Reviewer #1: Yes

Reviewer #2: Yes

4. Is the manuscript presented in an intelligible fashion and written in standard English?

Reviewer #1: Yes

Reviewer #2: Yes

 We are gratified the reviewers found our manuscript rigorous, clearly written, and technically sound and that our conclusions are well-supported by our data. We are also gratified the reviewers appreciate how our peel-1-DMS approach will increase the efficiency of plasmid-based CRISPR-Cas9 genome engineering in C. elegans by promoting screening-free genome editing at diverse loci. We found the reviewers comments insightful and constructive – especially by pointing out where more detailed methodological descriptions were needed and by providing us a potential approach to combine peel-1-DMS with cloning-free methods for CRISPR-Cas9 genome editing. We have added all requested methodological descriptions, replicate number clarifications, additional citations, and discussion points. We feel these changes have made for a more measured, accessible, and impactful manuscript. 

5. Review Comments to the Author

Reviewer #1: In this manuscript the authors use negative selection based on expression of peel-1 to eliminate array carrying animals in the selection procedure for CRISPR/Cas9 integrants. Differentiating between array carrying animals and true integrants is a time-consuming step in genome editing procedures that use plasmid-based repair templates, and the use of peel-1 should indeed help speed up the procedure.

The manuscript data appears sound, but the manuscript would benefit from the inclusion of more details:

- The DMS editing strategy should also be summarized and/or visualized better. I had to go back to the original publication to look up the details. For example, it is true that the brightness and consistency of GFP fluorescence can be used to differentiate integrants from arrays, but it is not clear at all why that is. A slightly extended summary of the method would help the reader of this manuscript.

- The injected plasmids are also presented with very little detail. They are available online of course, but it would be convenient to have at hand.

 We completely agree that additional methodological description of the peel-1-DMS strategy and relevant plasmids will increase the accessibility and impact of the manuscript. We have added the following paragraphs to the results section to address this:

(Italics denote new text)

“We used an optimized DMS genome editing strategy and guide selection tool (http://genome.sfu.ca/crispr/) to design programmed deletions at two separate loci, the uncharacterized genes F53B6.7 and F10E9.2 (Fig 1A) [1,2]. These deletions are predicted to result in null alleles. In this DMS strategy, guide RNAs are designed to induce a DNA double-strand break at the target locus (Fig 1A). These guides are injected along with Cas9 and a cocktail of plasmids including a homology directed repair template as well as pharyngeal (Pmyo-2::mCherry) and body wall muscle (Pmyo-3::mCherry) fluorescent co-injection markers (Fig 1A). The repair template consists of homology arms corresponding to the regions upstream and downstream of the cut site/intended edit site (e.g. in the case of a deletion the homology arm boundaries border the region to be deleted) as well as a dual-marker selection cassette housed between loxP sites (Fig 1A). The dual-marker selection cassette consists of a Pmyo-2::GFP pharyngeal fluorescent GFP marker and a Prps-27::neoR antibiotic (Neomycin/G418) resistance gene. Upon successful induction of a DNA double strand break by Cas9 the repair template is integrated at the target locus in a subset of animals (Fig 1A). If desired, the loxP sites also allow for optional excision of the selection cassette via a second round of injection of a plasmid encoding Cre recombinase for near scarless genome editing.

 To improve screening efficiency, the antibiotic Neomycin (G418) is applied for drug selection 24h after injection to kill virtually all non-transgenic F1 progeny (Fig 1B). However, this strategy produces both bona fide genome-edited integrants and unwanted animals harboring the injected plasmids as extrachromosomal arrays (Fig 1B, left). Both of these populations will be resistant to Neomycin, necessitating further screening. Because extrachromosomal arrays are often variably overexpressed different pharyngeal cells of array-carrying animals will express variable amounts of the pharyngeal GFP marker. Integrants, in contrast, will express only a single copy of GFP from each genome-edited locus in every pharyngeal cell. This allows for visual differentiation of genome-edited integrants from animals harboring arrays – array-carrying animals display bright and uneven GFP that is absent in some cells while integrants display even and dim GFP signal in every pharyngeal cell. A proportion of genome-edited integrants will also lose the body wall and pharyngeal muscle mCherry markers present in the extrachromosomal arrays, allowing for another level of visual selection against arrays (for an in depth description of the DMS approach see [2] and [1]). 

 While conceptually appealing, this scheme normally results in far more arrays than integrants, which can only be differentiated manually by screening for the presence of body wall and pharyngeal muscle mCherry markers and/or the brightness/consistency of GFP fluorophore expression in the pharynx. In practice the mCherry transgenes are often too dim to confidently visualize, leaving GFP as the only way to identify integrants, effectively reducing the process to searching for a dim needle in a variably bright haystack (Fig 1B). We hypothesized that the addition of peel-1 negative selection to the optimized DMS pipeline (referred to hereafter as peel-1-DMS) delivered on the 5th day following injection would kill arrays without killing genome-edited integrants, which by that point would have lost the toxic extrachromosomal array. Indeed, we observed that while there were several integrants that had survived selection on days 7 and 8 post-injection, heat shock induction of peel-1 killed arrays (Fig 1C).” 

- The paper could also benefit from a more extensive description of various numbers, like the number of animals injected, the number of plates, the number of injected animals per plate, and the number of line obtained. The numbers are there, but spread throughout the text, legends and methods.

 Again, we agree that including these additional descriptions will increase accessibility and impact of the manuscript. In addition to including the n in all figure captions and making the underlying data freely available online, we have modified the figures to indicate the number of plates used in each experiment directly in the relevant bars in a way characteristic of many C. elegans genome engineering papers so that it will be familiar to readers (e.g. similar to visualizations in Au et al., 2018 G3; Figure 4 from Dickinson et al., 2015, Genetics [3]; or McDiarmid et al., 2018 Disease Models & Mechanisms). We have also adjusted the figure caption textual description of numbers so that the group listing of n follows the way it is read in the figure (from left to right, top to bottom). 

Finally, the authors do not mention anywhere that peel-1 selection is not applicable for editing approaches that do not use plasmids as repair template.

 This is an important point to raise as cloning-free methods are a popular approach to introduce small edits in C. elegans. We have included this point and also the idea proposed by Reviewer #2 to include the Prps-27::neoR as an ~2KB linear repair template in a cloning-free method. With further optimizations, the Prps-27::neoR linear repair template could be inserted to confer Neomycin resistance at diverse loci without the need for repair template cloning while a standard peel-1 plasmid could be used to simultaneously kill arrays. We have included these points and references to cloning-free methods in the discussion.

“Our plasmid-based peel-1-DMS selection approach is particularly useful for large-scale genome edits due to the increased efficiency conferred by the integrated DMS cassette (e.g. replacing C. elegans genes with genes from other species, deleting entire open reading frames, or deleting several adjacent genes simultaneously) [1,4]. However, several methods based on linear DNA repair templates have recently been developed that allow for efficient generation of small edits (typically up to fluorophore-sized insertions) in C. elegans without the need for cloning [5–12]. While these approaches currently require manual screening/isolation and/or PCR validation to identify genome-edited animals and have lower efficiency for large edits, with further optimization, peel-1-DMS could be adapted to these strategies to allow cloning- and screening-free genome editing. In principle, this could be achieved through integration of the ~2 kilobase Prps-27::neoR Neomycin resistance sequence (e.g. via Nested CRISPR) [10] and simultaneous use of a peel-1 negative selection plasmid to kill array-carrying animals.”

Reviewer #2: Despite many germ nuclei can the targeted in a single injection with CRISPR reagents in C. elegans, the efficacy of CRISPR gene editing is still rather low, and therefore several methods have been developed to select animals carrying the edit of interest.

In this manuscript, the authors combine previous strategies to develop a new methodology.

In brief, to use resistance to antibiotics (in this case neomycin) as a marker of a CRISPR-Cas9 induced gene disruption, since the repair template is a plasmid, there is some residual activity from the no integrated plasmid that provides resistance to the antibiotic. To eliminate those worms carrying the extrachromosomal arrays made by plasmids, they add to the plasmid mix a plasmid expressing (upon heat shock) the toxin PEEL-1. In that way, they got plate enriched in worms with the right deletion (resistant to antibiotic because the insertion disrupting the gene, and not because resistance provided from the plasmid.

Strategies are not novel but it is a smart application of the existing tools. C. elegans is a pioneer in the applications of CRISPR technologies because of its genetic properties and short life cycle, and therefore any study on this matter would be of interest. The experiments are performed correctly and the manuscript is clear in its main messages. Thus, I think it would be suitable for publication in Plos One. 

 We are thankful to the reviewer for their positive assessment. 

Still, I have some comments and a few recommendations to improve the paper (mainly the discussion).

Producing mutations is very efficient in C. elegans with cloning-free CRISPR methods that would require not many PCRs to detect a mutant. In that sense, their approach is not of great interest to somebody with interest in producing a mutation by CRISPR. According to them, the interest of their methodology relies on “opening the door for genome-wide knock-out and phenotyping efforts” since the couple the CRISPR editing with an automated phenotyping device. Still, the main limitation for a large-scale experiment is the microinjection. They should mention this in the discussion and rather say that their approach would be valid to study gene families or a limited collection of genes, not for genome-wide approaches. Order two 500 bp gBlocks, plus cloning, would cost money and time…particularly if they go for large-scale approaches.

 The reviewer is correct that several cloning-free methods have recently been developed in C. elegans that are highly efficient for small edits or even insertions of fluorescent proteins. However, these methods typically require manual isolation of putative genome-edited animals based on a visible phenotype, and, in our hands, often require more PCR than advertised to identify bona fide genome-edited animals. Our peel-1-DMS approach, in contrast, allows for screening- and PCR-free isolation of genome-edited animals because they are the only animals both resistant to Neomycin and spared from peel-1 selection. We have also switched to using eBlocks which are much cheaper than gBlocks (eBlocks are currently listed at $0.08 CAD/bp) and ship faster (1-3 business days for eBlocks versus ~7 days for gBlocks). This makes the cost and reagent delivery time similar to (or in many cases less than) typical cloning-free methods. We have made the eBlock option clear in our methods. Our DMS approach is also highly efficient for large edits currently outside the feasible range of cloning-free methods (e.g. replacing entire C. elegans genes with tagged human genes in a single injection or deleting multiple adjacent genes simultaneously; McDiarmid et al., 2018, Disease Models & Mechanisms; Au et al., 2018 G3)). 

 However, we do agree that the discussion could be more measured. We have modified the discussion to reflect the fact that this approach is particularly useful for large edits, removed reference to the possibility of “genome-wide” investigations, and have added citation to several cloning-free methods. 

Order two 500 bp gBlocks, plus cloning, would cost money and time…particularly if they go for large-scale approaches. Such limitations would be bypassed with the insertion of a dsDNA with the rps-27 promoter plus the gene for neomycin resistance (I estimate 2Kb). This possibility should be discussed, and the articles from Dokshin et al 2018 (use Hybrid dsDNA donors) and Vicencio et al 2019 (Nested CRISPR) should be cited since they would provide a way to insert the 2Kb sequence that can provide resistance to the antibiotic and the time that a gene is mutated.

 This is an excellent suggestion! We agree that with additional optimizations it may soon be possible to employ peel-1-DMS using cloning-free methods, and that it is likely already possible to achieve Neomycin resistance via insertion of the ~2KB Prps-27::neoR sequence using existing methods. Expression of the Neomycin resistance gene from this linear repair template would likely still confer unwanted Neomycin resistance to animals who have not undergone genome editing, but this could be avoided using peel-1 negative selection against these array-carrying animals. We have included this exciting possibility of cloning- and screening-free CRISPR genome editing in the discussion. 

“Our plasmid-based peel-1-DMS selection approach is particularly useful for large-scale genome edits due to the increased efficiency conferred by the integrated DMS cassette (e.g. replacing C. elegans genes with genes from other species, deleting entire open reading frames, or deleting several adjacent genes simultaneously) [1,4]. However, several methods based on linear DNA repair templates have recently been developed that allow for efficient generation of small edits (typically up to fluorophore-sized insertions) in C. elegans without the need for cloning [5–12]. While these approaches currently require manual screening/isolation and/or PCR validation to identify genome-edited animals and have lower efficiency for large edits, with further optimization, peel-1-DMS could be adapted to these strategies to allow cloning- and screening-free genome editing. In principle, this could be achieved through integration of the ~2 kilobase Prps-27::neoR Neomycin resistance sequence (e.g. via Nested CRISPR) [10] and simultaneous use of a peel-1 negative selection plasmid to kill array-carrying animals.”

Also missing citation of Farboud et al, 2019, where authors comments on insertions of large fragments of DNA by CRISPR.

We have added this citation.

References

1. Au V, Li-Leger E, Raymant G, Flibotte S, Chen G, Martin K, et al. CRISPR/Cas9 Methodology for the Generation of Knockout Deletions in Caenorhabditis elegans. G3. 2018; g3.200778.2018. doi:10.1534/g3.118.200778

2. Norris AD, Kim H-M, Colaiácovo MP, Calarco JA. Efficient Genome Editing in Caenorhabditis elegans with a Toolkit of Dual-Marker Selection Cassettes. Genetics. 2015;201. 

3. Dickinson DJ, Pani AM, Heppert JK, Higgins CD, Goldstein B. Streamlined Genome Engineering with a Self-Excising Drug Selection Cassette. Genetics. 2015;200: 1035–1049. doi:10.1534/genetics.115.178335

4. McDiarmid TA, Au V, Loewen AD, Liang J, Mizumoto K, Moerman DG, et al. CRISPR-Cas9 human gene replacement and phenomic characterization in Caenorhabditis elegans to understand the functional conservation of human genes and decipher variants of uncertain significance. Dis Model Mech. 2018;11: dmm036517. doi:10.1242/dmm.036517

5. Chiu H, Schwartz HT, Antoshechkin I, Sternberg PW. Transgene-Free Genome Editing in Caenorhabditis elegans Using CRISPR-Cas. Genetics. 2013;195: 1167–1171. doi:10.1534/genetics.113.155879

6. Ward JD. Rapid and precise engineering of the Caenorhabditis elegans genome with lethal mutation co-conversion and inactivation of NHEJ repair. Genetics. 2014;199: 363–377. doi:10.1534/genetics.114.172361

7. Arribere JA, Bell RT, Fu BXH, Artiles KL, Hartman PS, Fire AZ. Efficient marker-free recovery of custom genetic modifications with CRISPR/Cas9 in Caenorhabditis elegans. Genetics. 2014;198: 837–846. doi:10.1534/genetics.114.169730

8. Kim H, Ishidate T, Ghanta KS, Seth M, Conte D, Shirayama M, et al. A Co-CRISPR strategy for efficient genome editing in Caenorhabditis elegans. Genetics. 2014;197: 1069–1080. doi:10.1534/genetics.114.166389

9. Dokshin GA, Ghanta KS, Piscopo KM, Mello CC. Robust genome editing with short single-stranded and long, partially single-stranded DNA donors in Caenorhabditis elegans. Genetics. 2018;210: 781–787. doi:10.1534/genetics.118.301532

10. Vicencio J, Martínez-Fernández C, Serrat X, Cerón J. Efficient generation of endogenous fluorescent reporters by nested CRISPR in Caenorhabditis elegans. Genetics. 2019;211: 1143–1154. doi:10.1534/genetics.119.301965

11. Farboud B, Severson AF, Meyer BJ. Strategies for efficient genome editing using CRISPR-Cas9. Genetics. 2019;211: 431–457. doi:10.1534/genetics.118.301775

12. Dickinson DJ, Goldstein B. CRISPR-Based Methods for Caenorhabditis elegans Genome Engineering. Genetics. 2016;202: 885–901. doi:10.1534/genetics.115.182162

6. PLOS authors have the option to publish the peer review history of their article (what does this mean?). If published, this will include your full peer review and any attached files.

Do you want your identity to be public for this peer review? For information about this choice, including consent withdrawal, please see our Privacy Policy.

Reviewer #1: No

Reviewer #2: No

---

## [Decision Letter · Decision Letter 1]

27 Aug 2020

Peel-1 negative selection promotes screening-free CRISPR-Cas9 genome editing in Caenorhabditis elegans.

PONE-D-20-16105R1

Dear Dr. Rankin,

We’re pleased to inform you that your manuscript has been judged scientifically suitable for publication and will be formally accepted for publication once it meets all outstanding technical requirements.

Kind regards,

Denis Dupuy, Ph.D.

Academic Editor

PLOS ONE

Additional Editor Comments (optional):

Reviewers' comments:

Reviewer's Responses to Questions

**Comments to the Author**

1. If the authors have adequately addressed your comments raised in a previous round of review and you feel that this manuscript is now acceptable for publication, you may indicate that here to bypass the “Comments to the Author” section, enter your conflict of interest statement in the “Confidential to Editor” section, and submit your "Accept" recommendation.

Reviewer #1: All comments have been addressed

Reviewer #2: All comments have been addressed

2. Is the manuscript technically sound, and do the data support the conclusions?

Reviewer #1: Yes

Reviewer #2: Yes

3. Has the statistical analysis been performed appropriately and rigorously? 

Reviewer #1: Yes

Reviewer #2: Yes

4. Have the authors made all data underlying the findings in their manuscript fully available?

Reviewer #1: Yes

Reviewer #2: Yes

5. Is the manuscript presented in an intelligible fashion and written in standard English?

Reviewer #1: Yes

Reviewer #2: Yes

6. Review Comments to the Author

Reviewer #1: The textual changes have improved the readability of the manuscript considerably, and have addressed my original criticisms.

Reviewer #2: Thanks for addressing my concerns and paying attention to my comments. Good luck for the next experiments!

7. PLOS authors have the option to publish the peer review history of their article (what does this mean?). If published, this will include your full peer review and any attached files.

Reviewer #1: No

Reviewer #2: **Yes: **Julián Cerón

---

## [Editor Report · Acceptance letter]

11 Sep 2020

PONE-D-20-16105R1 

*Peel-1* negative selection promotes screening-free CRISPR-Cas9 genome editing in *Caenorhabditis elegans*. 

Dear Dr. Rankin:

I'm pleased to inform you that your manuscript has been deemed suitable for publication in PLOS ONE. Congratulations! Your manuscript is now with our production department. 

Kind regards, 

on behalf of

Dr. Denis Dupuy 

Academic Editor

PLOS ONE